# Novel Acetamide-Based HO-1 Inhibitor Counteracts Glioblastoma Progression by Interfering with the Hypoxic–Angiogenic Pathway

**DOI:** 10.3390/ijms25105389

**Published:** 2024-05-15

**Authors:** Agata Grazia D’Amico, Grazia Maugeri, Luca Vanella, Valeria Consoli, Valeria Sorrenti, Francesca Bruno, Concetta Federico, Antonino Nicolò Fallica, Valeria Pittalà, Velia D’Agata

**Affiliations:** 1Department of Drug and Health Sciences, University of Catania, 95125 Catania, Italy; agata.damico@unict.it (A.G.D.); valeria_consoli@yahoo.it (V.C.); sorrenti@unict.it (V.S.); valeria.pittala@unict.it (V.P.); 2Department of Biomedical and Biotechnological Sciences, Section of Anatomy, Histology and Movement Sciences, University of Catania, 95100 Catania, Italy; 3Department of Biological, Geological and Environmental Sciences, Section of Animal Biology, University of Catania, 95123 Catania, Italyfederico@unict.it (C.F.); 4Department of Molecular Medicine, College of Medicine and Medical Sciences, Princess Al-Jawhara Centre for Molecular Medicine, Arabian Gulf University, Manama 329, Bahrain

**Keywords:** HO-1 gene, HO-1 inhibitors, glioblastoma multiforme, hypoxia, angiogenesis, vascular endothelial growth factor

## Abstract

Glioblastoma multiforme (GBM) represents the deadliest tumor among brain cancers. It is a solid tumor characterized by uncontrolled cell proliferation generating the hypoxic niches in the cancer core. By inducing the transcription of hypoxic inducible factor (HIF), hypoxia triggers many signaling cascades responsible for cancer progression and aggressiveness, including enhanced expression of vascular endothelial growth factor (VEGF) or antioxidant enzymes, such as heme oxygenase-1 (HO-1). The present work aimed to investigate the link between HO-1 expression and the hypoxic microenvironment of GBM by culturing two human glioblastoma cell lines (U87MG and A172) in the presence of a hypoxic mimetic agent, deferoxamine (DFX). By targeting hypoxia-induced HO-1, we have tested the effect of a novel acetamide-based HO-1 inhibitor (VP18/58) on GBM progression. Results have demonstrated that hypoxic conditions induced upregulation and nuclear expression of HO-1 in a cell-dependent manner related to malignant phenotype. Moreover, our data demonstrated that the HO-1 inhibitor counteracted GBM progression by modulating the HIFα/HO-1/VEGF signaling cascade in cancer cells bearing more malignant phenotypes.

## 1. Introduction

Glioblastoma multiforme (GBM) is classified as grade IV astrocytoma and represents the deadliest brain cancer, affecting adults with poor prognosis [1,2]. Based on a histological and molecular approach, the World Health Organization (WHO) has classified this type of glioma as glioblastoma isocitrate dehydrogenase (IDH)-wild type. The median survival of the affected patient is between 14 and 17 months due to cancer relapse. The difficulty in treating it is mainly due to the self-renewal ability of cancer cells that are responsible for tumor development, therapeutic resistance, and recurrence after treatment. To date, the gold standard therapy consists of a combined approach represented by surgery followed by radio- and chemotherapy [3]. This latter consists of a combined approach represented by surgery followed by radio- and chemotherapy [3]. The gold standard molecule used for pharmacological treatment is temozolomide (TMZ), which unfortunately shows many limitations as it is related to hematological and hepatic impairment [4,5]. Furthermore, TMZ is now sidelined due to increased chemoresistance phenomena mediated by Poly (ADP-ribose) Polymerase-1 (PARP-1) [6]. Comorbidities associated with chemotherapy as well as the recurrence rate of treated GBM highlight the demand to identify new glioma molecular targets able to improve/restore cancer cell sensitivity to standard therapy.

The uncontrolled cell proliferation characterizing GBM induces the formation of hypoxic niches inside the cancer core. The hypoxic microenvironment, by inducing transcription of hypoxia-inducible factors (HIFs), triggers the activation of different signaling cascades, making the tumor highly aggressive [7]. The pivotal role exerted by HIF activation has been summarized in many recent reviews that have highlighted its involvement in driving GBM progression [8,9,10,11].

More specifically, low oxygen tensions induce transcription of hypoxia-inducible factors (HIFs), including HIF-1α, that translocate into the nucleus, where it binds to constitutively expressed β subunit (HIF-1 β), forming an active complex. The latter, through the activation of hypoxia-responsive element (HRE), regulates many downstream target genes, including vascular endothelial growth factor (VEGF), which is the main factor responsible for aberrant neovascularization characterizing GBM progression. Besides activating the HIF1α–VEGF pathway, hypoxia also modulates gene expression of many other factors, with consequent upregulation of protective enzymes, such as human heme oxygenase 1 (HO-1) [12,13,14].

HO-1 is the enzyme responsible for endogenous heme degradation with consequent generation of three metabolites, including carbon monoxide (CO) and biliverdin [15], which exert anti-apoptotic, anti-inflammatory, and antiproliferative activity, resulting in an overall cytoprotective effect [16]. However, the role exerted by this enzyme is controversial and depends on the cellular microenvironment and the pathophysiological context in which it is expressed [17]. To address this concern, it has been demonstrated that HO-1 plays a pivotal role in regulating redox homeostasis, especially in cancer cells, which can generate excessive ROS as a result of abnormally rapid proliferation. HO-1 overexpression has been observed to be involved in the development of several types of cancer, including pancreatic and prostatic cancer, renal carcinoma, myeloma, and lung adenocarcinomas [18,19,20,21], and it has been widely associated with chemoresistance insurgence [22].

The promoter region of the HO-1 gene (HMOX1 gene) contains sites for various transcription factors, such as HIF-1α, nuclear factor erythroid 2–related factor 2 (Nrf2), and nuclear factor-kappa B (NF-KB) [23,24]. They can all be considered HO-1 transcription factors as they bind the HMOX1 promoter region under different conditions, including hypoxia and oxidative stress [25], making HO-1 part of the cellular response to hypoxia [15]. It has been reported that hypoxia stimulates oxidative stress with consequent ROS generation that in an Nrf2-dependent manner induces HO-1 expression [26,27]. Moreover, in vivo and in vitro studies have demonstrated that HO-1 upregulation under hypoxia is driven by HIF1α mediation [28,29]. In more detail, it has been demonstrated that the DNA coding sequence for HO-1 is localized downstream of HRE; therefore, their activation via HIF-1α stimulates the HO-1 promoter, resulting in enzyme-increased expression in vascular cells [30]. Recently, the HO-1 aberrant levels have also been associated with GBM stemness and invasion features [31]. In this tumor, mRNA encoding HO-1 expression has been shown to correlate with enhanced vascular density in high-grade gliomas because it also increases VEGF expression. Therefore, the accumulation of HO-1 was proposed as an indicator of neoangiogenesis [32,33,34].

It is worth noting that the upregulation of HO-1 reported in different human cancers induced suppression of apoptotic cell death through activation of mitogen-activated protein kinases (MAPK) pathways determining poor prognosis and chemoresistance [35]. Based on this evidence, it is possible to sustain that HO-1 plays an important role in the progression of GBM. Accordingly, Castruccio et al. 2019 [36], by using an in vitro model of GBM, have demonstrated the involvement of HO-1 in the regulation of cancer progression, showing that its overexpression is related to GBM cell proliferation and colony formation. Moreover, by comparing the transcriptome dataset of brain biopsies from different glioma grades, the authors have affirmed that HO-1 is overexpressed in human gliomas compared to non-malignant samples, demonstrating the oncogenic role of HO-1 during GBM development [36].

Taken together, all of this evidence suggests the inhibition of HO-1 as a possible antitumor strategy [37]. Accordingly, it has been demonstrated that HO-1 inhibition leads to antiproliferative activity in several tumoral cell lines [38,39] and tumor regression in several animal models [40,41,42,43]. Therefore, the identification of new specific enzymatic inhibitors able to target HO-1 activity could be useful in combined therapy. In line with this suggestion, our research group recently identified a class of HO-1 inhibitors showing antiproliferative activity toward GBM cells. The most interesting compound, named VP18/58 (molecular structure reported in Figure 1) [44], was able to reduce HO-1 enzymatic activity and the cell invasion rate by interfering with the angiogenesis in GBM cells. This compound showed good drug-likeness properties due to its favorable ADME (absorption, distribution, metabolism, and excretion) profile. This novel HO-1 inhibitor has also reduced invasion potential as well as VEGF expression in GBM cells, allowing us to suggest its involvement in counteracting different biological events underlying cancer progression [44].

Although the relationship among ROS, HIF1α, and HO-1 has been well-established in different cellular types [29,45,46], the impact of HO-1 and the hypoxic microenvironment has not yet been investigated in GBM. Therefore, in the present work, we aimed to investigate the role of HO-1 in hypoxia-driven GBM progression by testing the effect of the new synthesized azole-based HO-1 inhibitor VP18/58 on the hypoxia-triggered pathway. In this work, we have used two human glioblastoma cell lines, U87MG and A172, characterized by different tumorigenic potential [46], exposed to a hypoxic mimetic agent, deferoxamine (DFX). Overall, the present data showed a correlation between HO-1 overexpression and GBM cells bearing a malignant phenotype (U87MG). Furthermore, VP18/58 treatment counteracted GBM progression by interfering with the hypoxic–angiogenic pathway in U87MG cells.

## 2. Results

### 2.1. HO-1 Is Overexpressed in Human GBM Glioblastoma Cells Exposed to Hypoxia

To determine the correlation between HO-1 and hypoxia, we used two human GBM cell lines, U87MG and A172, possessing two different tumorigenic potentials [47]. The expression levels of HIF-1α, the main hypoxic transcription factor, have been evaluated at three different time points of 24 h, 48 h, and 72 h after DFX (100 μM) addition to the cells. As shown in Figure 2, HIF-1α was differently expressed in U87MG and A172 in basal conditions (control group). Based on the time course evaluation of HIF-1 α levels, and according to a previous investigation by [48], we chose to perform further experiments at 24 h as it represented the time point showing higher expression of HIF-1α in GBM cells (Figure 2 ** or *** vs. control).

To characterize the correlation between HO-1 and hypoxia, we performed the Western blot and immunofluorescence analysis of U87MG and A172 cells exposed to hypoxia for 24 h. The HO-1 protein expression was higher in DFX-treated U87MG cells than in the control group (*** vs. control) (Figure 3A,B), whereas its expression was unchanged in A172 cells (Figure 3C,D). To detect its cellular distribution, we performed an immunofluorescence analysis. Cell serial sections obtained through confocal laser scanning microscopy showed a predominant localization of HO-1 in the perinuclear area of the U87MG control group (Figure 3E). In contrast, DFX-induced hypoxia prompted its expression into the nucleus (Figure 3F). On the contrary, HO-1 immunoreactivity was faintly and exclusively expressed in the cytoplasm and perinuclear area of A172 cells in both experimental conditions (Figure 3G,H). The different results regarding HO-1 distribution could be ascribable to the specific cell genotypes as the U87MG overexpresses nestin and vimentin, two malignancy markers, whereas the A172 cells are less tumorigenic [47].

### 2.2. HO-1 Inhibition Affected GBM Cell Viability and Migration by Interfering with the Hypoxia-Driven Signaling Cascade

Based on the observed pike of HIF-1α at 24 h, we performed a dose-response curve in physiological conditions (normoxia) by treating GBM cells with VP18/58 for 24 h. As shown in Figure 4, cell viability was significantly reduced at a concentration of 10 μM in both cell lines (Figure 4 * or ** vs. control).

It is worth noting that when cells were exposed to hypoxic insult (DFX), the VP18/58 (10 μM) treatment reduced viability more in U87MG than in A172 cells (Figure 5, ^#^ or ^###^ vs. DFX).

To investigate the molecular mechanism underlying the effect of the HO-1 inhibitor on GBM cells, we tested its impact on the hypoxic–angiogenic pathway. As shown in Figure 6, the hypoxic microenvironment induced upregulation of HIF-1α, VEGF, and HO-1 in U87MG cells compared to the control, whereas their expression was significantly reduced after VP18/58 treatment (Figure 6A,C; ** or *** vs. control, ^###^ vs. DFX). In A172 cells, hypoxia enhanced HIF-1α and VEGF levels compared to the control group, but it did not affect HO-1 expression (Figure 6B,D; * or *** vs. control). The treatment with VP18/58 exclusively reduced HO-1 expression in both experimental conditions (Figure 6B,D; * or ** vs. control, ^##^ vs. DFX).

These data were corroborated by immunofluorescence analysis performed with confocal microscopy. The fluorescent signals for HO-1 (green fluorescence) and HIF-1α (red fluorescence) in DFX-cultured U87MG cells were largely detected in both nuclear and cytoplasmic compartments (Figure 7, Panel A). In contrast, HO-1 and HIF-1α were faintly expressed in the cytoplasm of A172 cells, where HO-1 immunoreactivity was further reduced following VP18/58 exogenous administration in both experimental groups. No difference in HIF-1α immunosignal was detected after VP18/58 treatment in this cell line.

To evaluate whether VP18/58 treatment affects GBM cell migration in the hypoxic condition, we performed a wound-healing assay in both cell lines exposed to DFX for 24 h with or without the HO-1 inhibitor. In line with our previous investigation [50], DFX treatment induced a significant increase in the number of cells moving into the wounded area with respect to their relative controls (Figure 8B,D *** vs. control). Contrariwise, the administration of VP18/58 to hypoxia-exposed cells reduced their migration rate more efficaciously in U87MG cells compared to A172 cells (Figure 8, ^#^ or ^###^ vs. DFX).

## 3. Discussion

Recent emerging evidence has reported aberrant levels of HO-1 in different human cancers, including GBM [31], whose overexpression is linked to a poor prognosis [35,51,52,53]. Although the conventional therapeutic approach is direct to provoke oxidative stress to promote cancer cell apoptosis [54,55], tumor cells respond to therapy by enhancing the antioxidant defense through HO-1 overexpression, therefore counteracting the efficacy of pharmacological treatment [25]. In line with the literature data [36], we reported HO-1’s different expression in the two human GBM cell lines U87MG and A172 after 24 h of hypoxia exposure. Accordingly, previous findings have also demonstrated that HO-1 was highly expressed under hypoxic conditions in colorectal cancer [56].

Recently, it has been suggested that a non-canonical function of HO-1 is directly linked to its nuclear translocation. More specifically, it has been demonstrated that HO-1 nuclear expression in hypoxic conditions is linked to chemoresistance or tumor progression in myeloid leukemia cells and human head and neck squamous cell carcinoma [57,58]. HO-1 nuclear expression under hypoxia is linked to antioxidant-responsive promoter activation as well as transcription factor induction, which in turn regulate HO-1 expression [24,59,60]. In the present investigation, we have reported significant evidence showing an increased HO-1 immunoreactivity at nuclear levels, especially in U87MG cells exposed to a hypoxic microenvironment; on the contrary, no nuclear translocation was detected in A172 cells characterized by a less malignant phenotype. These data suggest that HO-1 nuclear expression in hypoxic conditions is related to cancer malignancy.

Based on data present in the literature [22,61], we support here the existing hypothesis that HO-1’s nuclear expression is strictly linked to the regulation of gene transcription, including induction of HIF-1α, which promotes tumor progression. Thus, the HO-1 promoter, HMOX1, contains sites for various transcription factors activated under oxidative stress conditions, including HIF-1α, making HO-1 part of the cellular response to hypoxia [23,25]. It is known that HIF is an important metabolic regulator associated with glucose-6-phosphate dehydrogenase (G6PD) [62], but it is also known, based on literature findings on the correlation between HIF-1α and NAD(P)H Quinone Dehydrogenase 1 (NQO1), that increased expression of the transcriptional factor is observed to be accompanied by an increase of NQO1. Thus, it is plausible to conclude that the compound VP18/58’s inhibitory activity towards both HIF-α and HO-1 should reflect inhibition of downstream gene/protein expression of this factor together with pentose-phosphate-pathway-related genes [62,63,64,65].

The upregulation of HO-1 has been also related to the migratory abilities and metastasis formation of non-small-cell lung cancer, pancreatic cancer, and oral squamous cell carcinoma [66,67,68]. Therefore, the use of many pharmacological inhibitors of HO-1, including metalloporphyrins, modified protoporphyrins, or imidazole-based compounds, has been proposed to improve cancer cells’ response to conventional therapy [25]. Because competitive HO-1 inhibitors, such as metalloporphyrins, showed poor selectivity towards other heme-containing enzymes and HO-1 upstream induction phenomena, the medicinal chemistry research focused on the development of non-competitive HO-1 inhibitors, such as azole-based derivatives. As a consequence, there has been increasing attention paid to the identification and/or rational design of new non-porphyrinic HO-1 enzyme inhibitors [44,69,70]. Fallica et al. recently designed, synthesized, and tested a new battery of acetamide-based HO-1 inhibitors endowed with strong antiproliferative activity against a panel of tumoral cells, including lung, prostate, and GBM cells [44]. Among these, VP18/58 was selected because it was able to significantly reduce GBM cancer cell viability and counteract the enzymatic activity of HO-1. Based on these findings, in the current study, we have further characterized the molecular mechanism underlying VP18/58’s effects in GBM cells.

It is well-known that tumor invasiveness is strictly linked to the induction of the hypoxic–angiogenic pathway. In this regard, it has been reported that increased levels of HO-1 are paralleled with the increased levels of HIF-1α and induction of its downstream target gene, VEGF, in bladder cancer [71]. Moreover, it has been demonstrated that HO-1 inhibitor Zinc protoporphyrin was able to reduce colorectal cancer cell proliferation and migration by decreasing HIF-1α and VEGF levels [56]. In line with this, we have confirmed a correlation between HO-1, HIF-1α and VEGF in human GBM cells. These factors, overexpressed in GBM cells exposed to a hypoxic microenvironment [72,73], were significantly downregulated by VP18/58 treatment in a tumorigenic cell line (U87MG). Conversely, exogenous administration of VP18/58 to hypoxia-exposed A172 cells reduced HO-1 levels, whereas it did not exert any modulatory effect on HIF-1α and VEGF expression. These results were deepened by immunofluorescence analyses that have also revealed inhibition of HO-1 nuclear expression after VP18/58 treatment in cells bearing a more malignant phenotype, U87MG. This molecule also counteracted the migration rate of U87MG cells exposed to hypoxia. This result is supported by studies showing that HO-1 upregulation represents a molecular mechanism to maintain homeostasis by sustaining cell proliferation and migration [74,75]. Therefore, the strong inhibition of HO-1 could be directly responsible for lower cell invasiveness recorded in the DFX-treated group. VP18/58, being non-competitive and non-structurally related to the heme HO-1 inhibitor, shows different effects on protein expression. It is known that canonical inhibitors of HO catalytic activity, such as metalloporphyrin (e.g., SnMP, ZnPP, SnPP) [76], usually show an increasing effect on HO-1 protein expression; however, this behavior can give rise to opposite effects limiting these compounds’ clinical use. Here, we analyzed the effect of a novel inhibitor whose efficacy is probably correlated to its ability to reduce both the catalytic activity and protein expression of HO-1, which is found to be overexpressed in glioblastoma. We observed an interesting ability of VP18/58 to reduce HIF-1α expression; thus, we hypothesize that in vitro HO-1 protein reduction may be HIF-1α-dependent.

Although further studies are needed to better characterize the effects of the compound VP18/58, in the present investigation, we have demonstrated that this novel inhibitor exerts its activity by downregulating the hypoxic–angiogenic pathway in GBM, playing a crucial role in counteracting cell cancer migration depending on cell phenotype. Therefore, targeting HO-1 could be suggested as a novel strategy to improve cancer cell sensitivity to conventional pharmacological approaches.

## 4. Materials and Methods

### 4.1. Human GBM Cell Lines and Treatments

Human GBM cell lines, U87MG (cat. no. HTB-14) and A172 (cat. no. CRL-1620), were purchased from the American Type Culture Collection (ATCC). Cells were cultured as previously described [50]. To mimic hypoxia, we used a hypoxia-mimetic iron chelator, deferoxamine (DFX) (100 µM; Sigma-Aldrich, St. Louis, MO, USA). The use of exogenous administration of DFX, representing a hypoxia-mimetic iron chelator, offers the advantage of allowing the experimenter to open the culture plate or dish numerous times without altering the hypoxic conditions compared to the cell incubation method in the hypoxic chamber. Cells were treated with a novel acetamide-based HO-1 inhibitor named VP18/58 (10 μM), listed as 7l in a previous paper [44].

### 4.2. Cell Viability Assay

Cell viability was assessed using 3-[4,5-dimethylthiazol-2-yl]-2,5-diphenyltetrazolium bromide salt (MTT) (Sigma-Aldrich), as previously described [77]. Briefly, U87MG and A172 GBM cells were seeded in 96-well plates at a density of 1 × 10^4^ cells/well in 100 μL of culture medium for 24 h. Subsequently, cells were treated with eight different concentrations of VP18/58 (1 nM, 10 nM, 50 nM, 100 nM, 1 μM, 10 μM, 50 μM, 100 μM) for 24 h, and then the medium was replaced with a fresh medium with MTT salt added to each well for 3 h. Finally, dimethyl sulfoxide (DMSO) was used to dissolve formazan salts, and absorbance was measured at 570 nm using a microplate reader (Biotek Synergy-HT, Winooski, VT, USA). Six replicate wells were used for each group.

### 4.3. Wound-Healing Assay

Human GBM cells (U87MG and A172) were cultured in six-well dishes, and, after reaching the confluence, they were scratched with a 200 μL pipette tip. Each well was washed with PBS solution. Then, the cells were cultured in a medium containing DFX with or without VP18/58 compound for 24 h. A quantitative valuation of the wound area was executed as previously described [78]. Original microphotographs are reported in Appendix A.

### 4.4. Western Blot Analysis

Proteins were extracted by using RIPA buffer with a protease inhibitor cocktail (Roche Diagnostics, Basilea, Switzerland). The cell lysates’ protein concentration was calculated using the Quant-iT Protein Assay kit (cat. no. Q33211; Invitrogen, Carlsbad, CA, USA). About a 36 μg of each protein homogenate was processed to be separated through electrophoresis on Precast Protein Gels (cat. no. 4561084), as previously described [79]. The primary antibodies used were rabbit anti-HO-1 (1:200) (GeneTex, Irvine, CA, USA, GTX101147), mouse anti-HIF-1α (1:200) (NB 100-105), goat polyclonal anti-VEGF (1:200, cat. no. sc-1836), and rabbit polyclonal anti-β-actin (GeneTex GTX109639). The secondary antibodies used were goat anti-rabbit IRDye 800CW (1:20,000, cat. no. 926-32211) and goat anti-mouse IRDye 680CW (1:30,000, cat. no. 926-68020D; LI-COR Bio-sciences, Lincoln, NE, USA). The membranes, 1 h after secondary antibody incubation, were scanned with the Odyssey Infrared Imaging System, and ImageJ software (NIH, Bethesda, MD, USA; available at http://rsb.info.nih.gov/ij/index.html, accessed on 2 May 2024) was used to analyze blot density. The β-actin was used as a loading control. Original images are reported in Appendix A.

### 4.5. Immunofluorescence Assay

The human glioblastoma cell lines U87MG and A172 were cultured on glass coverslips and processed to perform immunofluorescence analysis as previously described by [80], allowing for the detection of HO-1 and HIF-1α distribution. The secondary antibodies used were Alexa Fluor 488-conjugated goat anti-rabbit (1:20,000; Catalog #A-11008, Life Technologies, Milano San Felice, Italy) or Alexa Fluor 594-conjugated goat anti-mouse (1:30,000; Catalog #A-21203, Life Technologies). Cell nuclei were stained with diamidino-2-phenylindole, DAPI (blue fluorescence). Immunolocalization was analyzed through confocal laser scanning microscopy (Zeiss LSM700, Oberkochen, Germany). All acquisitions were performed with ZEN-2010 software (https://zen-2010.software.informer.com/, accessed on 2 May 2024) (Zeiss Germany). Original microphotographs are reported in Appendix A.

### 4.6. Statistical Analysis

Data are reported as mean ± S.E.M. Statistical significance was assessed via an unpaired two-tailed Student’s *t*-test to compare the differences between two experimental groups or through One-Way Analysis of Variance (ANOVA) to compare differences among three or more groups, and statistical significance was assessed using the Tukey–Kramer post hoc test. The level of significance for all statistical tests was *p* ≤ 0.05.

## Figures and Tables

**Figure 1 ijms-25-05389-f001:**
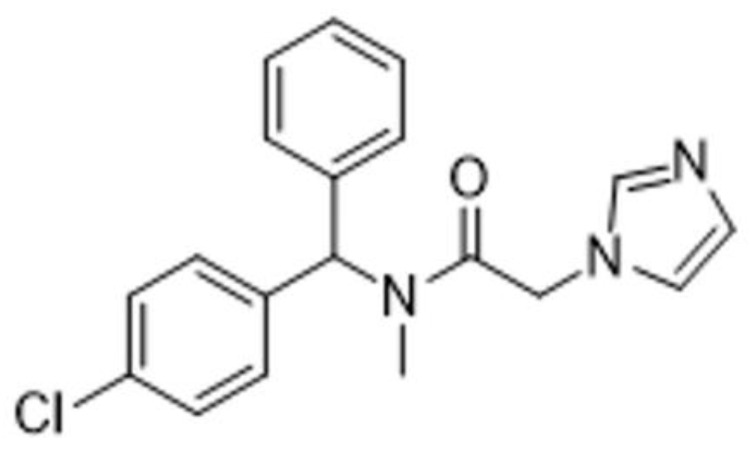
Chemical structure of VP18/58 [44].

**Figure 2 ijms-25-05389-f002:**
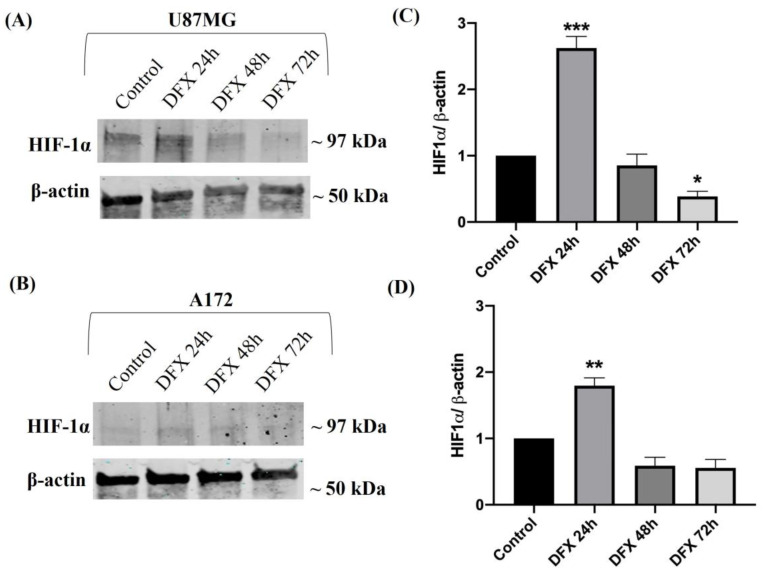
HIF-1α expression in human U87MG and A172 GBM cells exposed to hypoxia. (**A**,**B**) Representative immunoblots of signals detected by HIF-1α antibody obtained using cell homogenate from U87MG and A172 cultured in normoxia (control) and in DFX-induced hypoxia (DFX) at 24 h, 48 h, and 72 h. (**C**,**D**) The bar graphs show a quantitative analysis of signals obtained by three independent experiments. The ImageJ software was used to quantify the relative band density obtained by normalizing the protein levels to β-actin, representing a loading control. The values are expressed as the mean ± SEM by setting the control group value to 1 (* *p* < 0.05, ** *p* < 0.01, or *** *p* < 0.001 vs. control, as determined through One-Way ANOVA followed by Tukey’s post hoc test).

**Figure 3 ijms-25-05389-f003:**
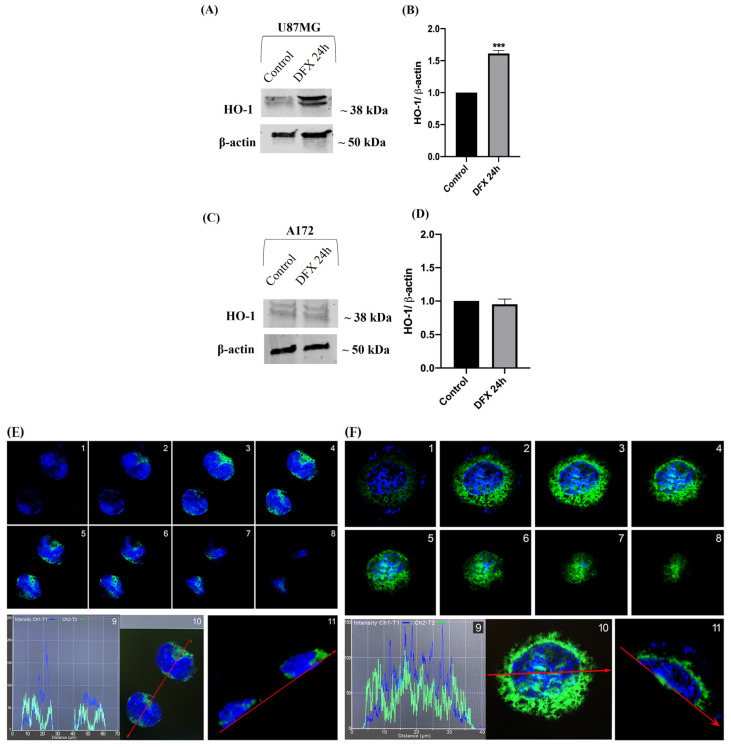
HO-1 expression and distribution in U87MG and A172 GBM cells exposed to hypoxia for 24 h. (**A**,**C**) Representative immunoblots of signals detected by HO-1 antibody obtained using cell homogenate from U87MG and A172 cells cultured in normoxia (control) and in DFX-induced hypoxia (DFX). (**B**,**D**) The bar graphs show a quantitative analysis of signals obtained in three independent experiments. The ImageJ software was used to quantify the relative band density obtained by normalizing the protein levels to β-actin, representing a loading control. The values are expressed as the mean ± SEM by setting the control group value to 1 (*** *p* < 0.001 vs. control, as determined by unpaired two-tailed Student’s *t*-test). (**E**,**G**) Photomicrographs show the immunofluorescence signal of HO-1 expression (green fluorescence) in U87MG (**E**) and A172 (**G**) cells cultured in normoxia and in DFX-induced hypoxia (**F**,**H**, respectively). (1–8): Serial section obtained through confocal laser scanning microscopy as previously described by [49]; each image represents a serial section along the *z*-axis of the same cell, with each section being 0.33 µm thick. Graph (9) shows the fluorescence intensity of HO-1 (green line) and DAPI (blue line) along the red line indicated in (10). (11) shows the 3D section along the red line in the same cells (scale bars, 10 μm).

**Figure 4 ijms-25-05389-f004:**
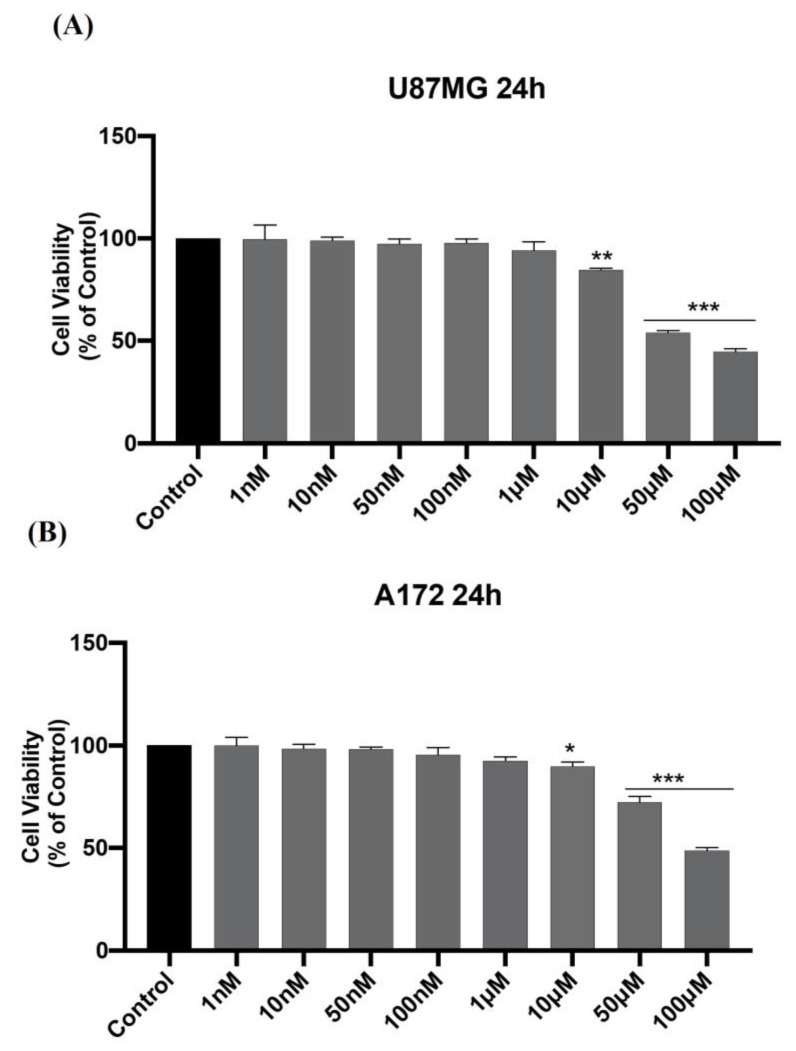
The dose–response curve of U87MG and A172 GBM cell viability following VP18/58 treatment. (**A**,**B**) Dose–response analysis of U87MG and A172 cell viability after 24 h of VP18/58 treatment (1 nM, 10 nM, 50 nM, 100 nM, 1 μM, 10 μM, 50 μM, 100 μM). The bar graphs show the results of three independent experiments, and the values are expressed as a percentage (%) of control (* *p* < 0.05, ** *p* < 0.01, or *** *p* < 0.001 vs. control, as determined through One-Way ANOVA followed by Tukey’s post hoc test).

**Figure 5 ijms-25-05389-f005:**
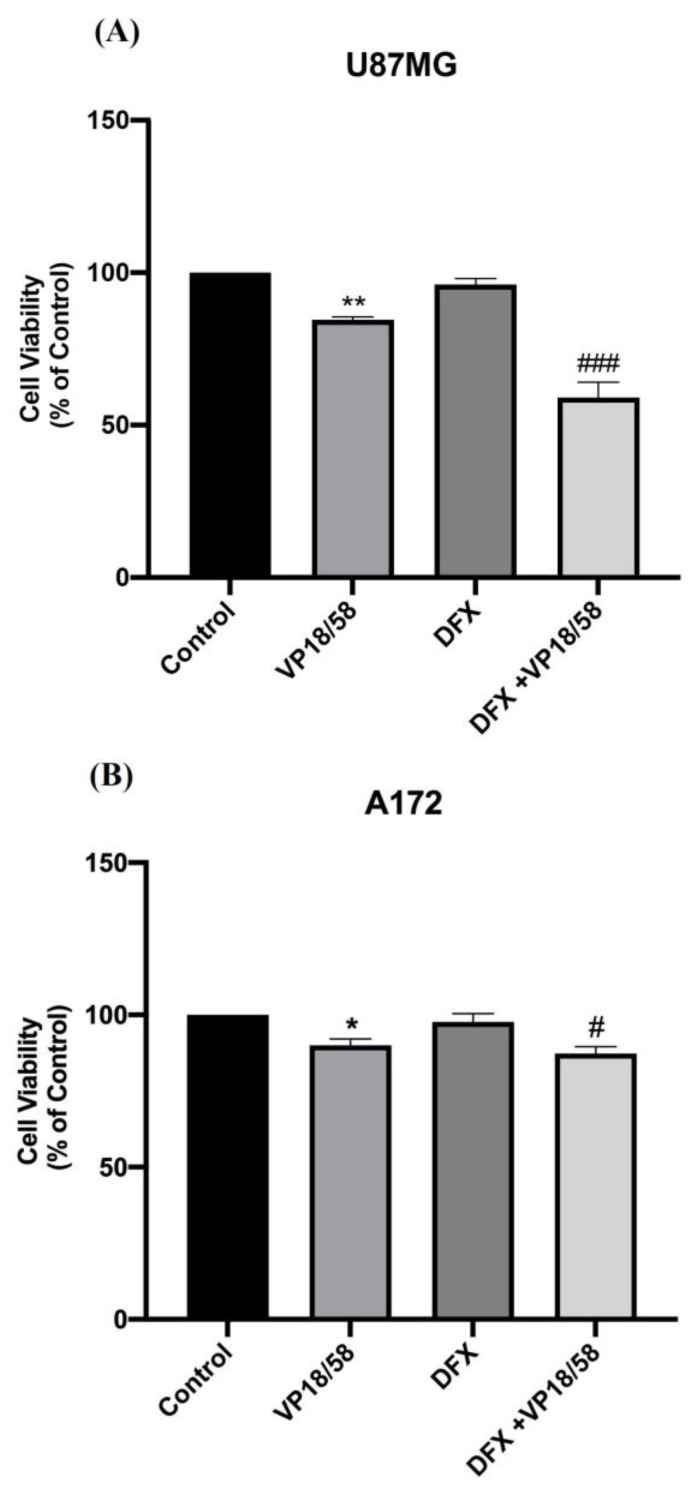
VP18/58 effect on U87MG and A172 cell viability exposed to hypoxia. (**A**,**B**) Effect of VP18/58 compound (10 μM) on U87MG and A172 cell viability exposed to hypoxic insult (DFX). The bar graphs show values expressed as a percentage (%) of control (* *p* < 0.05 or ** *p* < 0.01 vs. control, ^#^ *p* < 0.05 or ^###^ *p* < 0.001 vs. DFX, as determined through One-Way ANOVA followed by Tukey’s post hoc test).

**Figure 6 ijms-25-05389-f006:**
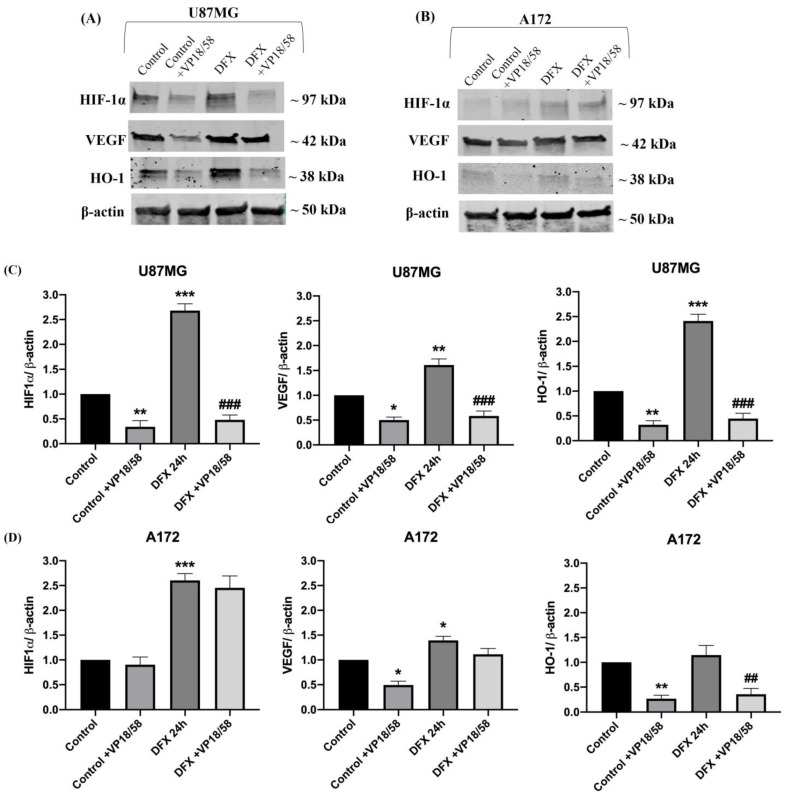
HIF-1α, VEGF, and HO-1 expression in U87MG and A172 cells exposed to hypoxic insult and treated with VP18/58. (**A**,**B**) Representative immunoblots of signals detected by HIF-1α, VEGF, and HO-1 antibodies obtained using cell homogenate from U87MG (**A**) and A172 (**B**) cells cultured in normoxia (control) and in DFX-induced hypoxia (DFX) for 24 h with or without VP18/58 compound (10 μM). (**C**,**D**) The bar graphs show the results of three independent experiments. The ImageJ software was used to quantify the relative band density obtained by normalizing the protein levels to β-actin, representing a loading control. The values are expressed as the mean ± SEM by setting the control group value to 1 (* *p* < 0.05, ** *p* < 0.01, or *** *p* < 0.001 vs. control, ^##^
*p* < 0.01 or ^###^
*p* < 0.001 vs. DFX, as determined through One-Way ANOVA followed by Tukey’s post hoc test).

**Figure 7 ijms-25-05389-f007:**
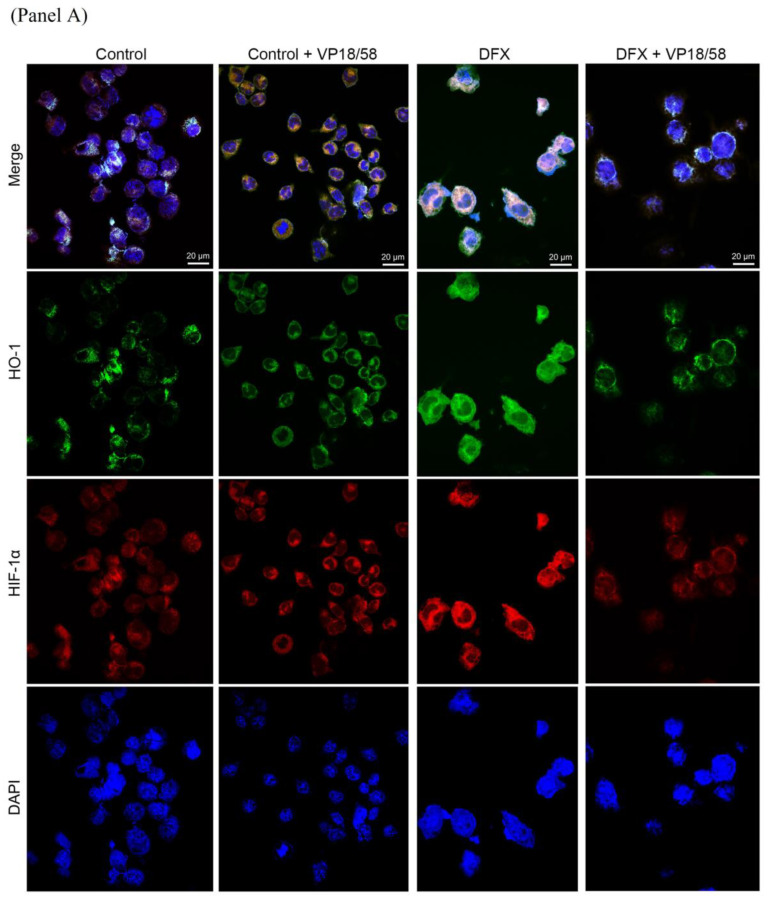
VP18/58 compound interferes with HO-1 and HIF-1α distribution in U87MG and A172 cells exposed to hypoxia. Photomicrographs show the immunofluorescence signal of HO-1 (green fluorescence) and HIF-1α expression (red fluorescence) in U87MG cells (**Panel A**) and A172 cells (**Panel B**) treated with 10 μM of VP18/58 showing either normoxia or DFX-induced hypoxia. Nuclei were stained with DAPI (blue fluorescence). Scale bar, 20 μm.

**Figure 8 ijms-25-05389-f008:**
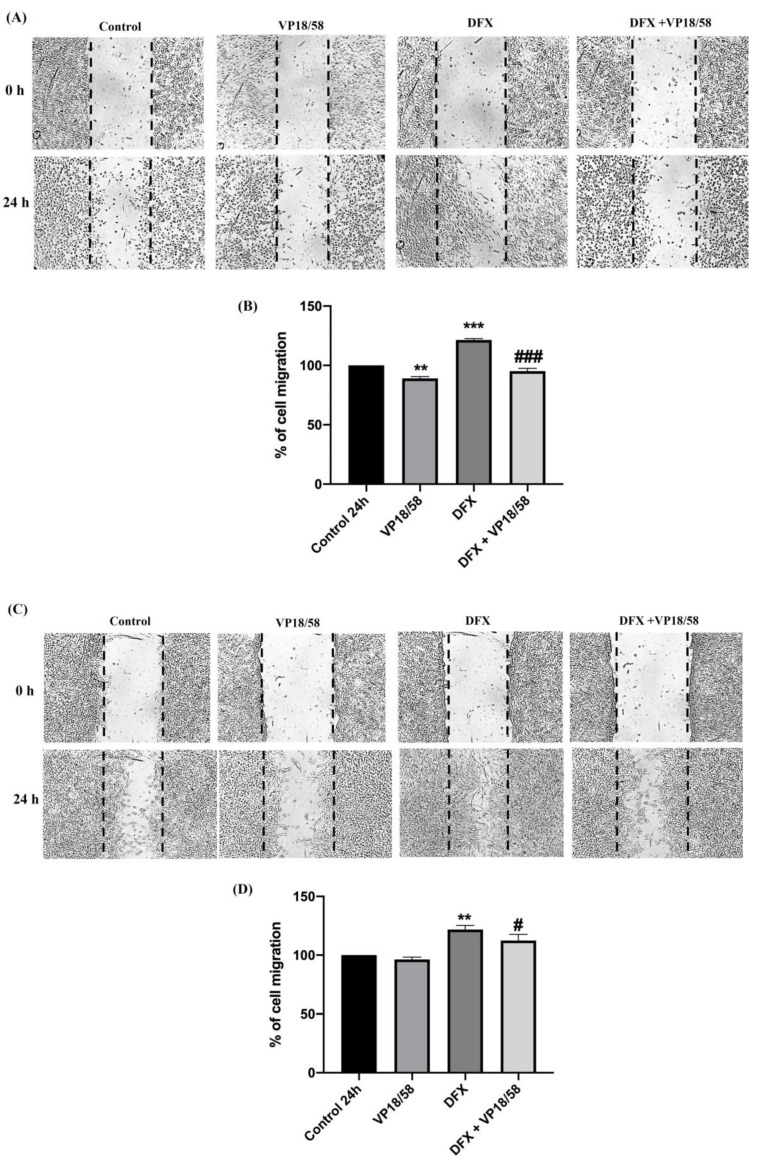
Effect of VP18/58 treatment on GBM cell migration following hypoxic insult. (**A**–**C**) Cell monolayers of U87MG (**A**) and A172 (**C**) cells were scraped with a pipette tip and incubated with the VP18/58 compound for 24 h. The wounded area was visualized under a microscope for quantification. Migration was calculated as the average number of cells observed in four random wounded fields per well in duplicate wells. (**B**–**D**) The bar graphs show values expressed as a percentage (%) of cell migration of U87MG (**B**) and A172 (**D**) compared to their relative controls. Data are represented as means ± standard error of the mean (SEM) (** *p* < 0.01 or *** *p* < 0.001 vs. control, ^#^ *p* < 0.05 or ^###^ *p* < 0.001 vs. DFX, as determined through One-Way ANOVA followed by Tukey’s post hoc test).

## Data Availability

Data is contained within the article and Appendix A.

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
