# Peer review of "Novel Acetamide-Based HO-1 Inhibitor Counteracts Glioblastoma Progression by Interfering with the Hypoxic–Angiogenic Pathway"

_ijms, 2024, doi:10.3390/ijms25105389_

Round 1
Reviewer 1 Report
Comments and Suggestions for Authors
The purpose of the presented work was to study the biological action of the new HO-1 inhibitor and to show its ability to counteract glioblastoma progression by interfering with the hypoxic-angiogenic pathway. The research demonstrates that in U87MG cells exposed to hypoxic conditions the expression of both HO-1 and HIF-1-alpha decreases in the presence of the investigated VP18/58 compound. The interaction with intracellular localization of HO-1 was also shown.
However, most of the conclusions regarding the mechanism of action of VP18/58 can be questioned due to the limited experimental data available.
Major concerns are as follows:
-
Other compounds from the non-porphyrin-based class of HO inhibitors, such as azalanstat, were not investigated in parallel with V18/58. If similar phenomena were observed upon inhibition of HO-1 by other inhibitors, it would serve as confirmation of the role of HO-1 and the mechanism of V18/58 action. Further, in a previous study by the authors (https://doi.org/10.1021/acs.jmedchem.1c00633), a series of compounds inhibiting heme oxygenase was synthesized and investigated. It would be interesting to compare the effect of VP18/58 (previously listed as 7l), with compound 7n, which much more selectively inhibits inducible HO-1 compared to constitutive HO-2 (IC50 = 1 uM vs. 46 uM).
-
The authors speculate that HO-1 inhibition affects cell viability and migration by downregulating the hypoxic-angiogenic pathway. However, they did not explore any downstream participants of the HO-1 pathway. For example, it is known that induction of the nuclear HO-1 level leads to upregulation of NQO1 and G6PDH. It would be worthwhile to evaluate the expression levels of these or other genes associated with HO-1 expression and localization.
-
No visible increase in the expression of HIF-1-alpha and HO-1 was observed when A127 cells were exposed to the DFX compound at a concentration of 100 μM. Meanwhile, in the research of other authors, the induction of HIF-1 expression in A172 cells in the presence of DFX was much more noticeable (eg, Fig.2B in 10.1016/j.bbrc.2009.07.138 ). Thus, it cannot be reliably concluded that the hypoxic state was accurately simulated for A127 cell line. All conclusions drawn based on this information should be questioned.
-
According to the data presented, inhibition of the catalytic activity of HO-1 leads to a decrease in its expression on protein level. This observation is not discussed in the article. What mechanism of such regulation could be proposed?
-
It would have been worthwhile to test shorter incubation times because published data suggest that HIF-1alpha activation peaks between 4 and 10 hours after the induction of oxidative stress.
Other, less significant, comments are provided below:
-
Lines 35-36: Diagnosis ‘glioblastoma multiforme’ is outdated. According to the latest WHO CNS5 classification, this type of gliomas is called ‘Glioblastoma, IDH-wildtype’.
-
Throughout the text, it is worth clarifying under what conditions specific data were obtained (normoxia or hypoxia). For example, lines 177-178.
-
It is claimed that all western blots were conducted in triplicates. However, only one immunoblot is provided for each experiment in the supplementary file with the original images. It is necessary to provide photographs of all blots, as the narrow distribution of the standard error of the mean raises doubts.
-
Figure 8: The photographs of the wells with cells in normoxia with VP18/58 (A and C) are missing, although it is present in the histograms (B and D).
-
The scratch test photographs are provided on a very small scale. Photographs on a larger scale should be provided to visually assess the state of the cells. It was shown that compound VP18/58 at 10μM concentration exhibits cytotoxicity towards glioblastoma cells in normoxia and hypoxia (Figure 4 and 5). The seeming invasiveness reduction observed when culturing cells with VP18/58 and DFX may be caused by their cytotoxic action.
-
The scratch test photographs are provided on a too small scale to visually assess the state of the cells. As it was shown that compound VP18/58 at 10μM concentration exhibits cytotoxicity towards glioblastoma cells in normoxia and hypoxia (Figure 4 and 5), the seeming invasiveness reduction may be caused by DFX+VP18/58 cytotoxic action.
-
It would be beneficial to include higher-resolution original images obtained during the scratch test in the Supplementary file.
The article also exhibits some structural shortcomings. Information more appropriate for the introduction is found within the discussion section, while certain details that should belong in the materials and methods section are presented in the results. Additionally, there are errors and omissions. Specifically,
-
Lines 103-106: In my opinion, insufficient attention has been paid to the selection and description of the compound VP18/58. This information, found in the discussion section, particularly lines 270-286, would be more beneficial if relocated to the introduction section.
-
Lines 161-174: This information should be moved to the Materials and Methods section. Scale bars should be indicated on the figure or in the figure caption.
-
Line 217: Сonclusions regarding Figure 7 are missing - it is definitely incomplete (“In contrast, HO-1 and HIF-1α were faintly expressed in A172 cells, where with DFX+VP18/58.”)
-
Lines 123-124: It is necessary to specify that the starting point of 14, 48 and 72 h intervals is the addition of 100 uM DFX to the cells.
-
Figure 8: Cell lines are not specified. The description for Fig. 8C and 8D are missing in the figure caption.
Additional aspect not covered in the draft: do the cell lines U87MG and A172 exhibit differences in their levels of HO-2 expression? Can the basal level of HO-2 expression mitigate the effects of oxidative stress?
Comments on the Quality of English LanguageI did not find any serious problems with English language.
Author Response
Major concerns are as follows:
Comment 1. Other compounds from the non-porphyrin-based class of HO inhibitors, such as azalanstat, were not investigated in parallel with V18/58. If similar phenomena were observed upon inhibition of HO-1 by other inhibitors, it would serve as confirmation of the role of HO-1 and the mechanism of V18/58 action. Further, in a previous study by the authors (https://doi.org/10.1021/acs.jmedchem.1c00633), a series of compounds inhibiting heme oxygenase was synthesized and investigated. It would be interesting to compare the effect of VP18/58 (previously listed as 7l), with compound 7n, which much more selectively inhibits inducible HO-1 compared to constitutive HO-2 (IC50 = 1 uM vs. 46 uM).
Replay to Comment 1. Thanks to the Reviewer for suggesting to investigate in parallel with V18/58 other compounds such as azalanstat. In the present work, we focused our investigation exclusively on the characterization of the V18/58 molecule. In the future investigation, we will attempt to deepen the molecular mechanism underlying V18/58 action also by comparing its effect with other non-porphyrin-based HO inhibitors.
As correctly reported by the Reviewer, in our previous investigation we have identified a class of HO-1 inhibitors showing antiproliferative activity toward GBM cells. Among these, we have selected exclusively VP18/58 (previously listed as 7l), since it was the most active compound against GBM in our previous work.
Comment 2. The authors speculate that HO-1 inhibition affects cell viability and migration by downregulating the hypoxic-angiogenic pathway. However, they did not explore any downstream participants of the HO-1 pathway. For example, it is known that induction of the nuclear HO-1 level leads to upregulation of NQO1 and G6PDH. It would be worthwhile to evaluate the expression levels of these or other genes associated with HO-1 expression and localization.
Replay to Comment 2. Thanks to the Reviewer for this notable suggestion. In our next investigation, we will attempt to explore this aspect by analyzing the downstream participants of the HO-1 pathway.
To emphasize the point raised by the Reviewer, we have included a sentence in lines 276-283 of the Discussion section as follows: “It is known that HIF is an important metabolic regulator associated with glucose-6-phosphate dehydrogenase (G6PD) [70] but it is also known, based on literature findings on the correlation between HIF-1α and NAD(P)H Quinone Dehydrogenase 1 (NQO1), that increased expression of the transcriptional factor is observed to be accompanied with also an increase of NQO1, thus it can be plausible to conclude that the compound VP18/58 inhibitory activity towards both HIF- α and HO-1 should reflect inhibition of downstream gene/ protein expression of this factor together with pentose phosphate pathway-related genes [70-73]”.
The added references were also included in the Reference list.
Comment 3. No visible increase in the expression of HIF-1-alpha and HO-1 was observed when A127 cells were exposed to the DFX compound at a concentration of 100 μM. Meanwhile, in the research of other authors, the induction of HIF-1 expression in A172 cells in the presence of DFX was much more noticeable (eg, Fig.2B in 10.1016/j.bbrc.2009.07.138). Thus, it cannot be reliably concluded that the hypoxic state was accurately simulated for A127 cell line. All conclusions drawn based on this information should be questioned.
Replay to Comment 3. In the Results section, Figure 2 panel B, the expression of HIF-1-alpha increased when A172 cells were exposed to DFX (100 μM) for 24h. Therefore, this result follows a similar trend to data reported in Fig.2B in 10.1016/j.bbrc.2009.07.138.
To better emphasize this point, we have added the suggested reference in the main text which was also added a sentence in line 140 of the Results section as follows: “… and according to a previous investigation by [55], …”
The added reference was also included in the Reference list.
Comment 4. According to the data presented, inhibition of the catalytic activity of HO-1 leads to a decrease in its expression on protein level. This observation is not discussed in the article. What mechanism of such regulation could be proposed?
Replay to Comment 4. Thanks to the Reviewer for this acute observation. To emphasize the suggested point, we have added a sentence in lines 317-326 of the Discussion section as follows: “VP18/58 being a non-competitive and non-structurally related to heme HO-1 inhibitor shows different effects on protein expression. It is known that canonical inhibitors of HO catalytic activity as metalloporphyrin (e.g. SnMP, ZnPP, SnPP) [84] usually show an increasing effect on HO-1 protein expression, however, this behavior can give rise to opposite effects limiting these compounds’ clinical use. Here we analyzed the effect of a novel inhibitor whose efficacy is probably correlated to its ability to reduce both catalytic activity and protein expression of HO-1 which is found to be overexpressed in glioblastoma. We observed an interesting ability of VP18/58 in reducing HIF-1α expression thus we hypothesize that in vitro HO-1 protein reduction may be HIF-1α-dependent.”
The added reference was also included in the Reference list.
Comment 5. It would have been worthwhile to test shorter incubation times because published data suggest that HIF-1alpha activation peaks between 4 and 10 hours after the induction of oxidative stress.
Replay to Comment 5.
Thank you to the reviewer for the precious suggestion. We are planning to test the effect of VP18/58 on the HIF-1α pathway during shorter incubation times.
Comment 6. Lines 35-36: Diagnosis ‘glioblastoma multiforme’ is outdated. According to the latest WHO CNS5 classification, this type of gliomas is called ‘Glioblastoma, IDH-wildtype’.
Replay to Comment 6. Following the Reviewer's suggestion, we have changed the sentence "glioblastoma multiforme" to "glioblastoma (GBM)" in the Manuscript. Moreover, to better clarify this point, we have included a sentence in lines 36-38 of the Introduction section as follows: “Based on a histological and molecular approach, the World Health Organization (WHO) has classified this type of glioma as Glioblastoma, isocitrate dehydrogenase (IDH)-wildtype [3-5].”
The added references were also included in the Reference list.
Comment 7. Throughout the text, it is worth clarifying under what conditions specific data were obtained (normoxia or hypoxia). For example, lines 177-178.
Replay to Comment 7. To clarify this point we have changed the sentence in lines 184-185 of the Results section as follows: “… we performed a dose-response curve in physiological conditions (normoxia) by treating GBM cells with VP18/58 for 24h.”
Comment 8. It is claimed that all western blots were conducted in triplicates. However, only one immunoblot is provided for each experiment in the supplementary file with the original images. It is necessary to provide photographs of all blots, as the narrow distribution of the standard error of the mean raises doubts.
Replay to Comment 8. Thanks to the reviewer for the comment. As requested by the reviewer, the blot replicates are reported below. To confirm the results, another investigator has performed further densitometric analyses. The data were confirmed and new bar graphs were added to Figures 2, 3, and 6. As requested in the instructions for authors, we only included in the supplementary file (S1) the uncropped blots of the representative blots shown in the manuscript.

Comment 9. Figure 8: The photographs of the wells with cells in normoxia with VP18/58 (A and C) are missing, although it is present in the histograms (B and D).
Replay to Comment 9. Many thanks to the reviewer for the right observation. We added in the manuscript the correct panel, which includes the photographs of the cells cultured in normoxia, and treated with VP18/58 corresponding to the results indicated in the bar graphs (Figure 8B and D).
Comment 10. The scratch test photographs are provided on a very small scale. Photographs on a larger scale should be provided to assess the state of the cells visually. It was shown that compound VP18/58 at 10μM concentration exhibits cytotoxicity towards glioblastoma cells in normoxia and hypoxia (Figures 4 and 5). The seeming invasiveness reduction observed when culturing cells with VP18/58 and DFX may be caused by their cytotoxic action.
Comment 11. The scratch test photographs are provided on a too small scale to assess the state of the cells visually. As it was shown that compound VP18/58 at 10μM concentration exhibits cytotoxicity towards glioblastoma cells in normoxia and hypoxia (Figure 4 and 5), the seeming invasiveness reduction may be caused by DFX+VP18/58 cytotoxic action.
Replay to Comments 10 and 11. Unfortunately, we only acquired the photographs of the scratch test at 10x magnification. The results of this test are in accord with the MTT assay, demonstrating that 10μM of VP18/58 treatment reduced cell viability by about 20%.
Comment 12. It would be beneficial to include higher-resolution original images obtained during the scratch test in the Supplementary file.
Replay to Comment 12. Following the Reviewer's suggestion, we have provided in a Supplementary file (S2) a Figure including higher-resolution original images obtained during the scratch test.
Comment 13. Lines 103-106: In my opinion, insufficient attention has been paid to the selection and description of the compound VP18/58. This information, found in the discussion section, particularly lines 270-286, would be more beneficial if relocated to the introduction section.
Replay to Comment 13. Following the Reviewer's suggestion, information about the description of the compound VP18/58 has been relocated in lines 114-118 of the Introduction section as follows: “This compound showed good drug-likeness properties due to its favorable ADME (absorption, distribution, metabolism, and excretion) profile. This novel HO-1 inhibitor has also reduced invasion potential as well as VEGF expression in GBM cells, allowing us to suggest its involvement in counteracting different biological events underlying cancer progression [51]”.
Comment 14. Lines 161-174: This information should be moved to the Materials and Methods section. Scale bars should be indicated on the figure or in the figure caption.
Replay to Comment 14. Following the Reviewer's suggestion, the indicated period was merged in the Materials and Methods section, and the scale bar was included in the caption of Figure 3.
Comment 15. Line 217: Сonclusions regarding Figure 7 are missing - it is incomplete (“In contrast, HO-1 and HIF-1α were faintly expressed in A172 cells, where with DFX+VP18/58.”)
Replay to Comment 15. Many thanks to the reviewer for the right observation. The mistyping was corrected in lines 224-227 of the Results section as follows: “In contrast, HO-1 and HIF-1α were faintly expressed in the cytoplasm of A172 cells, where HO-1 immunoreactivity was further reduced following VP18/58 exogenous administration in both experimental groups. No difference in HIF-1α immunosignal was detected after VP18/58 treatment in this cell line”.
Comment 16. Lines 123-124: It is necessary to specify that the starting point of 14, 48 and 72 h intervals is the addition of 100 uM DFX to the cells.
Replay to Comment 16. Following the Reviewer's suggestion, this specification was added in line 137 of the Results section as follows: “The expression levels of HIF-1α, the main hypoxic transcription factor, have been evaluated at three different time points 24h, 48h, and 72h after DFX (100 μM) addiction to the cells”.
Comment 17. Figure 8: Cell lines are not specified. The description for Fig. 8C and 8D are missing in the figure caption.
Replay to Comment 17. The description for Figure 8 (C and D) was included in lines 242-247 of the figure caption as follows: “A-C) Cell monolayer of U87MG (A) and A172 (C) cells were scraped by a pipette tip and incubated with VP18/58 compound for 24h. …….. B-D) The bar graphs show values expressed as a percentage (%) of cell migration of U87MG (B) and A172 (D) as compared to their relative controls.”
Comment 18. Additional aspect not covered in the draft: do the cell lines U87MG and A172 exhibit differences in their levels of HO-2 expression? Can the basal level of HO-2 expression mitigate the effects of oxidative stress?
Replay to Comment 18. It is an interesting observation, and we thank the reviewer for arising the matter. Indeed, HO-2 represents the constitutive isoform of the enzyme, being usually expressed indistinctly in both healthy and cancerous tissues. Thus, targeting HO-2 lack clinical significance, or rather it can be considered an undesired effect. It is known that HO-2 is responsible for most of HO activity, whilst HO-1 as inducible isoform is activated only under certain circumstances, as a response to stimuli like hypoxia, inflammation and oxidative stress (PMID 18289071-32455831- 33923744). In this context, HO-1 was intended to be the focus of the present work as primary target of the newly designed compound.
Reviewer 2 Report
Comments and Suggestions for Authors
The article in case "Novel acetamide-based HO-1 inhibitor counteracts glioblastoma progression by interfering with the hypoxic-angiogenic pathway" is a complex molecular approach to GBM future treatment. The authors used immunofluorescence assay, western-blot analysis together with cell viability assays on 2 GBM cell lines in order to determine impact of HO-1 and hypoxic microenvironment. The results clearly prove a positive association between HO-1 nuclear expression in hypoxic conditions and cancer malignancy. Hence targeting HO-1 is suggested as a novel strategy to improve cancer cell sensitivity to conventional pharmacological approaches.
Comments on the Quality of English LanguageSome lines need to be rephrased for better clarity.
e.g. line 39: This latter...
line 49: ...is the main responsible....
Author Response
Comment 1. e.g. line 39: This latter...
Replay to Comment 1. The indicated sentence was rephrased in line 41 of the Introduction section as follows: “To date, the gold standard therapy consists of a combined approach represented by surgery followed by radio and chemotherapy”
Comment 2. line 49: ...is the main responsible....
Replay to Comment 2. The indicated sentence was rephrased in lines 53-54 of the Introduction section as follows: “The hypoxic microenvironment, by inducing transcription of hypoxia‑inducible factors (HIFs), triggers the activation of different signaling cascades making the tumor highly aggressive [7]”
Reviewer 3 Report
Comments and Suggestions for Authors
Authors present an in-vitro study on human GBM cells to investigate Novel acetamide-based HO-1 inhibitor and its interractions with hypoxic-angiogenic pathway. Microenvironmental hypoxia is the main reason for activating signaling cascades which lead to higher agression of the tumor. Molecular biologist is needed to evaluate materials and methods of the study. Neovascularization of GBMs is a very important topic and authors demonstrate in their in-vitro study that it is possible to possibly counteract GBM progression by interfering its pathways. The title is somewhat misleading - proposed mechanisms can possibly counteract GBM progression, i.e. influence its angiogenesis, since progression the tumor has several factors. Include this article and comment:
Fallica AN, Sorrenti V, D'Amico AG, Salerno L, Romeo G, Intagliata S, Consoli V, Floresta G, Rescifina A, D'Agata V, Vanella L, Pittalà V. Discovery of Novel Acetamide-Based Heme Oxygenase-1 Inhibitors with Potent In Vitro Antiproliferative Activity. J Med Chem. 2021 Sep 23;64(18):13373-13393. doi: 10.1021/acs.jmedchem.1c00633. Epub 2021 Sep 2. PMID: 34472337; PMCID: PMC8474116. Include literature review on HIF in GBM treatment in form of a table.Author Response
Comment 1. Include this article and comment: Fallica AN, Sorrenti V, D'Amico AG, Salerno L, Romeo G, Intagliata S, Consoli V, Floresta G, Rescifina A, D'Agata V, Vanella L, Pittalà V. Discovery of Novel Acetamide-Based Heme Oxygenase-1 Inhibitors with Potent In Vitro Antiproliferative Activity. J Med Chem. 2021 Sep 23;64(18):13373-13393. doi: 10.1021/acs.jmedchem.1c00633. Epub 2021 Sep 2. PMID: 34472337; PMCID: PMC8474116.
Replay to Comment 1. The suggested article is included in the main text and also in the References list (number 50).
Comment 2. Include literature review on HIF in GBM treatment in form of a table.
Replay to Comment 2. Thanks to the Reviewer for this suggestion. Amazing recent Reviews have already summarized the HIF involvement in GBM. Therefore, to emphasize this point, in lines 55-57 of the Introduction section the following sentence was added: “The pivotal role exerted by HIF activation has been summarized in many recent reviews that have highlighted its involvement in driving GBM progression [8-11]”
The added references were also included in the Reference list.